# A Charcot-Marie-Tooth-Causing Mutation in HSPB1 Decreases Cell Adaptation to Repeated Stress by Disrupting Autophagic Clearance of Misfolded Proteins

**DOI:** 10.3390/cells11182886

**Published:** 2022-09-15

**Authors:** Xuelian Zhang, Yaru Qiao, Ronglin Han, Yingjie Gao, Xun Yang, Ying Zhang, Ying Wan, Wei Yu, Xianchao Pan, Juan Xing

**Affiliations:** 1Department of Pathophysiology, School of Basic Medical Science, Southwest Medical University, Luzhou 646000, China; 2Department of Medicine Chemistry, School of Pharmacy, Southwest Medical University, Luzhou 646000, China; 3State Key Laboratory of Genetic Engineering, School of Life Sciences, Zhongshan Hospital, Fudan University, Shanghai 200438, China

**Keywords:** Charcot-Marie-Tooth, HSPB1, repeated stress, misfolded proteins, axonal transport, autophagy

## Abstract

Charcot-Marie-Tooth (CMT) disease is the most common inherited neurodegenerative disorder with selective degeneration of peripheral nerves. Despite advances in identifying CMT-causing genes, the underlying molecular mechanism, particularly of selective degeneration of peripheral neurons remains to be elucidated. Since peripheral neurons are sensitive to multiple stresses, we hypothesized that daily repeated stress might be an essential contributor to the selective degeneration of peripheral neurons induced by CMT-causing mutations. Here, we mainly focused on the biological effects of the dominant missense mutation (S135F) in the 27-kDa small heat-shock protein HSPB1 under repeated heat shock. HSPB1^S135F^ presented hyperactive binding to both α-tubulin and acetylated α-tubulin during repeated heat shock when compared with the wild type. The aberrant interactions with tubulin prevented microtubule-based transport of heat shock-induced misfolded proteins for the formation of perinuclear aggresomes. Furthermore, the transport of autophagosomes along microtubules was also blocked. These results indicate that the autophagy pathway was disrupted, leading to an accumulation of ubiquitinated protein aggregates and a significant decrease in cell adaptation to repeated stress. Our findings provide novel insights into the molecular mechanisms of HSPB1^S135F^-induced selective degeneration of peripheral neurons and perspectives for targeting autophagy as a promising therapeutic strategy for CMT neuropathy.

## 1. Introduction

Charcot-Marie-Tooth disease (CMT; also known as hereditary motor and sensory neuropathy) is one of the most common inherited neurodegenerative disorders, characterized by profound distal muscle weakness and sensory deficits that progresses slowly in a length dependent manner [1]. CMT refers to a clinically heterogeneous group of inherited peripheral neuropathies with a broad range of phenotypes, inheritance patterns and causative genes. Up to date, approximately 90 causative genes have been linked to CMT [2]. Progressive and selective degeneration, predominantly of peripheral neurons, is the main pathological feature in patients with CMT [3]. However, how CMT-associated mutations exclusively affect the peripheral nervous system (PNS) remains an open question, the answer to which is crucial for better understanding the pathogenesis of CMT.

Dominant mutations in the gene encoding 27-kDa small heat-shock protein B1 (HSPB1, also known as HSP27) have been identified as causative for CMT type 2F (CMT2F, OMIM 606595) [4]. HSPB1 is a ubiquitously expressed chaperone protein that can bind misfolded or denatured proteins to prevent their aggregation under heat, hypoxia and oxidative stress, thus having a major protective role in cells [5,6]. In addition, HSPB1 mediates survival of damaged neurons by promoting axonal regeneration in neural system [7]. Therefore, dysfunction of HSPB1 is causatively associated with neurodegenerative disorders with exceptional frequency. Currently, at least 18 missense mutations in HSPB1 are known to cause CMT neuropathy [8]. Among them, the mutation of S135F in HSPB1 is frequently mentioned because it can cause both axonal CMT2F and distal hereditary motor neuropathy (distal HMN) [4,9]. Although a number of potential pathomechanisms underlying HSPB1 mutations such as the impairment of autophagy, altered chaperone activity, the tendency to aggregate, and axonal transport deficits have been proposed [10], little knowledge exists of the molecular mechanism for the HSPB1^S135F^-induced selective axon loss in PNS.

Axonal transport of organelle and proteins along microtubules is essential for axonal activity and neuron survival [11]. Emerging evidence suggests that anomalous protein interactions and axonal transport deficits are associated with mutant HSPB1-induced neuropathies [4,12,13]. By using transgenic mouse models, d’Ydewalle et al. found that defective axonal transport of mitochondria was shown in dorsal root ganglion (DRG) neurons isolated from 10-month-old (symptomatic) HSPB1^S135F^-expressing mice, mainly owing to a marked decrease in acetylated α-tubulin level in peripheral nerves [14]. In contrast, the mitochondrial transport was unaffected in peripheral neurons isolated from presymptomatic mutant HSPB1^S135F^-expressing mice. In later work, Almeida-Souza et al. demonstrated that enhanced binding of HSPB1^S135F^ to α-tubulin led to increased stability and reduced dynamics of microtubules in peripheral nerves from 3-month-old (presymptomatic) HSPB1^S135F^-expressing mice, however, with no change in tubulin acetylation [15]. They explained that CMT symptoms might be triggered by the transition of the microtubule network from a presymptomatic hyperstable state to a symptomatic unstable state [15,16]. These observations suggested that disturbances in microtubule dynamics are highly related to HSPB1^S135F^-induced CMT, but the underlying cause of selective degeneration of peripheral neurons still remains to be revealed.

The PNS is located outside the brain and spinal cord and is responsible for sensing external stimuli and transmitting motor and sensory information to the central nervous system (CNS). In contrast to the CNS protected by the bone of the spine and skull, the PNS is particularly vulnerable to stress stimuli exerted by daily life [17]. Of note, under repeated stress, excessive protein aggregates are formed by misfolded proteins [18], which can be degraded by ubiquitin-proteasome system (UPS) and autophagy [19]. Autophagic degradation heavily relies on microtubule-based axonal transport [11]. Microtubules are composed of polymerized tubulin, which can serve as the track for dynein motors mediating the transportation of misfolded proteins from the cytoplasm to the perinuclear region [20,21,22]. Therefore, we hypothesized that daily repeated stress might play a key role in driving the progressive and selective degeneration of PNS in CMT patients. Previously, Almeida-Souza et al. [13] found that HSPB1^S135F^ displayed higher chaperone activity and enhanced affinity to client proteins compared with the wild type. Weeks et al. [23] pointed out that the modified self-assembly of HSPB1^S135F^ and its abnormal interaction with partner proteins may affect normal function of HSPB1 in the cells. Unfortunately, the biological effects of the hyperactivation of HSPB1^S135F^ on the degeneration of peripheral neurons were not thoroughly investigated.

In this study, we investigated the effects of HSPB1^S135F^ on accumulating ubiquitinated protein aggregates under repeated heat shock, particularly focusing on the autophagy pathway. We found that HSPB1^S135F^ presented hyperactive binding to both α-tubulin and acetylated α-tubulin during heat shock when compared with the wild-type protein. The aberrant interactions with tubulin impeded the microtubule-based transport of ubiquitinated proteins and autophagosomes toward perinuclear region for the degradation by autophagy, resulting in the cytoplasmic accumulation of protein aggregates and the decrease of cell adaptation to repeated stress. These findings provide an insightful understanding of the molecular mechanisms of HSPB1^S135F^-induced CMT neuropathy.

## 2. Materials and Methods

### 2.1. Cells Lines and Primary Antibodies

SH-SY5Y, a human neuroblastoma cell line, was purchased from Weskong Company Limited (Chengdu, China). The antibodies against α-tubulin and actin were purchased from Invitrogen (Carlsbad, CA, USA). Anti-acetylated tubulin (K40) was purchased from Proteintech (Chicago, IL, USA). Antibodies against p62 and ubiquitin were purchased from Santa Cruz Biotechnolog (Santa Cruz, CA, USA). Antibodies against HDAC6 and Flag-tag were purchased from Abcam (Cambridge, UK).

### 2.2. Plasmid Construction and Cell Transfection

For overexpressing in HEK293T cells, the wild-type human HSPB1 gene (HSPB1^WT^) was cloned into the pRK7-Flag vector. S135F mutation of HSPB1 (HSPB1^S135F^) was generated by site-directed mutagenesis (Toyobo KOD Mut Kit). pRK7-Flag vector was used as control. Plasmids were identified by gene sequencing and amplified in Escherichia coli. Cell transfection of HEK293T was carried out by Lipo2000 (Invitrogen) according to the manufacturer’s protocol after reaching 50–70% cell confluence.

To construct HSPB1-expressing SH-SY5Y cells, we cloned Flag-tagged HSPB1^WT^ or HSPB1^S135F^ into the pLVX-puro vector. Subsequently, Flag-tagged HSPB1^WT^ or HSPB1^S135F^ was packed into lentivirus by transfecting pLVX-puro vector and Lenti-Mix (pMDLg/pRRE:pVSV-G:pRSV-Rev = 0.5:0.3:0.2) into HEK293T. With the help of lentivirus transfection, HSPB1^WT^ or HSPB1^S135F^ gene can be integrated into the SH-SY5Y genome. Transfected cells were screened by adding 2 ug/mL puromycin into the cell culture system.

### 2.3. Cell Culture and Repeated Thermal Stimulation

HEK293T and SH-SY5Y cells were cultured in DMEM high glucose medium (Gibco, Grand Island, NY, USA) supplemented with 10% fetal bovine serum (Gibco), 1% penicillin–streptomycin (Gibco). Cells were cultivated in a humidified atmosphere of 5% CO_2_ at 37 °C.

At 24 h post-transfection of HEK293T or 48 h post-seeding of SH-SY5Y, cells were exposed to repeat thermal stimulation. In brief, cells were heat-shock in a water bath at 42 °C for 1 h, denoted as the “heat (H)” group. Then, some H cells were put back to 37 °C for recovery for 3 h, followed by another heat-shock treatment for 1 h, denoted as the “heat-heat (HH)” group. Last, some HH cells were put back to 37 °C again for recovery for 6 h, denoted as the “heat-heat-recovery (HHR)” group. Cells that were always kept at the 37 °C incubators were employed as non-heat-shocked (untreated) controls.

To increase the stimulation intensity, short recovery time of 1 h between heat-shocks was adapted. As shown in Figure 1, the H cells were sequentially treated with recovery for 1 h, heat-shock for 1 h (denoted as “increased heat-heat, IHH”), recovery for 1 h, another heat-shock for 1 h (denoted as “increased heat-heat-heat, IHHH”), recovery for another 6 h (denoted as “increased heat-heat-heat-recovery, IHHHR”).

### 2.4. Western Blotting

At each time point described above, cells were gently washed in PBS three times. Then, total protein extraction was performed by lysing the cells with Radio Immunoprecipitation Assay (RIPA) buffer containing protease inhibitor. Then, equal amounts of protein were loaded onto 5–15% SDS-PAGE for separation and transferred to a nitrocellulose filter (NC) membrane. The membranes were then blocked with milk blocking buffer (Cwbiotech, Beijing, China) for 1 h at room temperature (r.t.) under gentle rocking and then incubated with primary antibodies overnight at 4 °C. The membranes were washed and incubated with appropriate secondary antibodies conjugated with IRDye (LI-COR, Lincoln, NE, USA) at r.t. for 1 h, followed by imaging using ODYSSEY CLx (LI-COR).

### 2.5. Co-Immunoprecipitation Assay

For co-immunoprecipitation, cells were lysed by 1% Nonidet P-40 (NP40) lysis buffer containing 50 mM Tris-HCl, pH 7.5, 150 mM NaCl, and protease inhibitors (aprotinin, leupeptin, pepstatin, Na_3_VO_4_, PMSF, NAM, TSA). Then, cell lysates were incubated with a Flag bead (Sigma Aldrich, St. Louis, MO, USA) with constant agitation overnight at 4 °C. The beads were washed six times with NP40 buffer and one additional time with 3×Flag peptide TBS buffer (50 mM Tris, 150 mM NaCl, pH 7.4). Then, the elution was performed by incubation the Flag beads for 1 h at 4 °C with 250 ng/L of the 3×FLAG peptide. Finally, the eluates were boiled with SDS loading buffer and subjected to Western blotting.

### 2.6. Immunofluorescence Assay

Immunofluorescence assay was performed according to our previous studies [24]. Briefly, cells were gently washed with PBS, followed by fixed in 4% paraformaldehyde for 30 min and permeabilized with 0.25% Triton X-100 for 10 min at SDS Blocking was performed with 5% BSA diluted in PBS for 1 h. The cells were than incubated with appropriate primary antibody overnight at 4 °C, washed three times in PBS, followed by incubation with secondary antibody (conjugated with Alexa-488 or Alexa-594) for 60 min at r.t. Finally, nucleus was stained with DAPI. The stained specimens were washed, mounted in glycerol and examined by a confocal laser scanning microscope (Leica, Wetzlar, Germany). 

### 2.7. Statistical Analysis

Experiments were performed in triplicate at least unless otherwise stated. Image analysis was carried out using Image J software (https://imagej.net/). Data analysis was performed using Origin 8.1 software (OriginLab, Northampton, MA, USA). Data are presented as mean ± standard deviation (SD). 

## 3. Results

### 3.1. HSPB1^S135F^ Binds Both to α-Tubulin and to Acetylated α-Tubulin

Previous studies have suggested that the binding of HSPB1 to α-tubulin and the reduced level of acetylated α-tubulin might be associated with the development of CMT [14,15]. To further investigate the expression of HSPB1 and its interaction with tubulin under stress conditions, western blot, co-immunoprecipitation and immunofluorescence staining were carried out for HEK293T and SH-SY5Y cells. Up-regulation of HSPB1 expression in response to cellular stress was observed in the previous study [25]. Here, a remarkable increase of HSPB1^WT^ and HSPB1^S135F^ expression was also observed in heat-shocked HEK293T cells (Figure 2A,B), confirming the important role of HSPB1 under stress conditions. Furthermore, co-immunoprecipitation revealed that HSPB1^S135F^ but not HSPB1^WT^ bound to both α-tubulin and acetylated α-tubulin (Figure 2A). Besides, heat-shock stress had no significant influence on the interaction between HSPB1^S135F^ and α-tubulin. To visualize the localization of wild type and mutant HSPB1 and their interactions with α-tubulin, SH-SY5Y cells were co-stained with antibodies of Flag tag and α-tubulin. In non-heat-shocked (untreated) cells, we observed that HSPB1^S135F^ and HSPB1^WT^ showed a similar distribution in the cytoplasm (Figure 2B). After heat shock, however, nuclear translocation of HSPB1^WT^ but not HSPB1^S135F^ was observed. Furthermore, HSPB1^S135F^ showed preferred co-localization with α-tubulin (Figure 2C,D). These results illustrated that HSPB1^S135F^ but not HSPB1^WT^ presents high affinity to α-tubulin and alteration in nuclear translocation, which might disturb their normal function and affect cell adaptability to repeated stress. 

### 3.2. HSPB1^S135F^ Blocks the Formation of Perinuclear Aggresomes under Repeated Heat Shock

Stress-induced misfolded proteins can be ubiquitinated and then transported to the perinuclear region along microtubules to form aggresomes, which can be subsequently degraded via the autophagic pathway [26]. A previous study has shown that histone deacetylase 6 (HDAC6), a microtubule-associated deacetylase, is required for aggresome formation [27]. Independent of its deacetylase activity, HDAC6 binds to misfolded proteins and recruits them to aggresomes around nucleus [28]. Besides its recruiting role, HDAC6 is also a component of aggresomes [22]. Accordingly, intracellular expression and distribution of HDAC6 can reflect aggresome formation under stress conditions. 

MG132 is a UPS inhibitor that can provoke aggresome formation [29]. Herein, cells treated with 10 µM of MG132 for 3 h were employed as the positive control. Results showed that the expression of HDAC6 was increased by MG132 (Figure 3A), confirming the activation of aggresome formation under stress conditions. For heat-shocked cells, significant up-regulation of HDAC6 expression was also observed. To investigate the functional effect of increased HDAC6, we first evaluated the acetylation levels of tubulin. As shown in Figure 3B, there was no significant difference in acetylated α-tubulin levels among all groups, consistent with the previous observation [15]. These results implied that the main function of heat shock-induced HDAC6 was to handle misfolded proteins. 

The intracellular distribution of HDAC6 was measured by immunofluorescence analysis (Figure 3C). The heat shock-induced perinuclear accumulation of HDAC6 was observed in vector or HSPB1^WT^-expressing cells, especially in “HH” cells wherein HDAC6 was found to agglomerate into punctate or linear aggregates. By contrast, only a few large diffused aggregates were observed in HSPB1^S135F^-expressing cells. A similar observation was also obtained in SH-SY5Y cells (Figure 3D). The distribution of intracellular HDAC6 suggested that the translocation of misfolded proteins from cell edge to perinuclear region was suppressed in HSPB1^S135F^-expressing cells, and the functional effect of HSPB1^S135F^ is independent of cell types. Furthermore, absence of juxtanuclear punctate structures around nucleus demonstrated that HSPB1^S135F^ blocked aggresome formation under stress conditions.

According to Kawaguchi’s research [22], HDAC6-enriched aggresomes can be easily discernible as a dark inclusion body under a phase-contrast microscope. In this work, aggresome formation in SH-SY5Y cells treated with MG-132 or “increased heat shock” was investigated. As shown in Figure 4, perinuclear dark inclusions were observed in stressed vector and HSPB1^WT^ cells, indicating the successful formation of aggresomes. The reduction in quantity and volume of inclusions after cell recovery suggested that stress-induced perinuclear aggresomes can be further degraded. In MG132 treated HSPB1^S135F^-expressing cells, although some black dot structures scattered in cells, they failed to form dark inclusions in the perinuclear region, suggesting the impaired formation of aggresomes. More interestingly, although the black dots seemed to be transported toward the perinuclear region in heat-shocked HSPB1^S135F^-expressing cells, they cannot be further degraded. The accumulation of dark dots around nucleus in “IHHH” HSPB1^S135F^-expressing cells is likely attributed to cell contraction under heat shock. These results further confirmed that the stress-induced formation of perinuclear aggresomes was inhibited by the HSPB1^S135F^ isoform. Impaired formation of aggresomes was also presented in HEK293T cells (Appendix A). 

### 3.3. HSPB1^S135F^ Inhibits the Degradation of Autophagosome under Repeated Heat Shock

In the autophagy pathway, ubiquitinated proteins can also be delivered by autophagosomes along microtubules. In this degradation process, misfolded proteins are first packaged into autophagosomes, which are double-membrane vesicles distributed in the cytoplasm. Then, autophagosomes are directed to move along microtubules toward the perinuclear region and fuse with lysosomes. Subsequently, the packaged proteins within autophagosomes are digested by lysosomal hydrolases [30]. Despite being different from aggresomes, protein degradation via autophagosomes also relies on microtubule transport. The autophagic receptor p62 (also known as sequestosome1, SQSTM1) has an essential role in recognizing ubiquitinated proteins and serves as an adaptor in autophagosomes [29]. To test whether the enhanced binding of HSPB1^S135F^ with α-tubulin affects autophagosomal movement along microtubules, the location and expression of p62 were detected. Not surprisingly, intracellular accumulation of p62 in heat-shocked cells was observed by immunofluorescence analysis (Figure 5A), highlighting a critical role of stress stimuli in the activation of autophagy. In stressed vector and HSPB1^WT^-expressing cells, p62 punctuate inclusions were formed adjacent to nucleus (Figure 5B). After 6 h of recovery, cellular cytoplasm became clean, especially in HSPB1^WT^-expressing cells, indicating accelerated clearance of p62 punctuate inclusions. By contrast, p62 punctuate inclusions had always distributed in the cytoplasm of heat-shocked HSPB1^S135F^-expressing cells, even in recovered cells (Figure 5B). More importantly, HSPB1^S135F^ seems to exacerbate intracellular accumulation of p62 as insoluble protein aggregates, indicated by the increased number of p62 punctuate inclusions in recovered cells that calculated using Image J after applying a fixed threshold to all pictures (Figure 5C). This suggested that HSPB1^S135F^ prevented the perinuclear transportation and degradation of p62 punctate inclusions, which was further confirmed by western blot (Figure 5D,E). The cytoplasmic accumulation of p62 was also observed in SH-SY5Y cells with HSPB1^S135F^ isoform (Appendix A). These results indicated that the microtubule-based transport of misfolded proteins via autophagosomes in autophagy pathway was damaged by the enhanced binding of HSPB1^S135F^ to tubulin.

### 3.4. HSPB1^S135F^ Prevents the Clearance of Misfolded Proteins

To further investigate the contribution of aggresomes and autophagosomes to the clearance of misfolded proteins, we tested the content of ubiquitinated proteins in untreated and heat-shocked HEK293T cells. As shown in Figure 6, a significant increase in protein ubiquitination was detected in heat-shocked cells. In “HH” group cells, ubiquitinated proteins decreased after recovery, indicating that the recovery time of 6 h is sufficient for cells to clear these proteins. For cells treated with increased heat shock (IHHH), recovery time longer than 6 h is required for cells to degrade stress-induced ubiquitinated proteins. For both “HH” and “IHHH” group cells, however, HSPB1^S135F^ obviously prevented the clearance of heat-induced ubiquitinated proteins when compared with HSPB1^WT^, especially for the proteins with high molecular weight. This confirmed the negative effect of HSPB^S135F^ in the turnover of misfolded proteins caused by repeated stress. 

### 3.5. HSPB1^S135F^ Decreases Cell Recovery from Repeated Heat Shock

To explore whether the defect in the clearance of misfolded proteins in HSPB1^S135F^-overexpressing cells affects cell recovery from repeated heat shock, cell survival state and morphologies were further analyzed. An inverted phase-contrast microscope can be employed to observe cell survival state [31]. As shown in Figure 5A, HSPB1^S135F^ increased the amount of apoptotic HEK293T cells in recovery groups, confirming the cytotoxicity of misfolded proteins demonstrated in the previous study [22]. Herein, the cell area and aspect ratio of SH-SY5Y cells were measured for cell state analysis (Figure 7B,C). Obvious heat-induced morphological changes were observed for HSPB1^WT^ and HSPB1^S135F^-expressing cells, as manifested by a reduction in aspect ratio and cell area, respectively. After cell recovery of 24 h, most of the HSPB1^WT^-expressing cells can return to their original aspect ratio (indicated by blue boxes in Figure 7C). In contrast, most HSPB1^S135F^-expressing cells sustained the state of heat-induced poor cell spreading (indicated by green boxes in Figure 7C), indicating their failure in recovery from repeated stress. These results clearly demonstrated that HSPB1^S135F^ has a deleterious effect on cell recovery from repeated stress.

## 4. Discussion

Dominant mutations in the small heat shock protein HSPB1 are causative for CMT neuropathy. Notably, the S135F mutation is the only one that causes both CMT2F and distal HMN by affecting mainly the peripheral nervous system [4]. Therefore, the pathogenesis of HSPB1^S135F^-induced CMT has recently attracted much interest. However, HSPB1 is a ubiquitously expressed molecular chaperone that participates in diverse physiological processes, such as thermotolerance, cytoskeletal dynamics, cell differentiation, apoptosis, and protein folding [9]. Thus, pinpointing the underlying mechanism for selective degeneration of peripheral neurons induced by mutations in HSPB1 is a great challenge. 

Neurons have long axons and large expanses of dendritic, which makes them sensitive to external stimuli [32]. Prominently, the peripheral neurons are chronically exposed to a broad of stressors in daily life. In physiological condition, axonal regeneration of PNS is much better than CNS, suggestive of a more robust adaptability to chronic stress [33]. Repeated exposure to various stressors could lead to an increasing number of misfolded proteins in the axons and nerve terminals of peripheral neurons. By using primary patient fibroblasts, Ylikallio et al. [34] found that ablation of the entire C-terminus of HSPB1 increased cellular sensitivity to protein misfolding and impaired the function of HSPB1 in cellular stress response. Therefore, it is plausible that HSPB1^S135F^ leads to reduced adaptation of peripheral neurons to daily repeated stress, largely due to the accumulation of misfolded proteins. However, to the best of our knowledge, the molecular mechanism of HSPB1^S135F^-induced accumulation of protein aggregates under repeated stress has been rarely reported.

Accumulation of ubiquitinated protein aggregates is a well-established cause of progressive neurodegeneration [35]. HSPB1 is a typical chaperone that can bind misfolded proteins and protect them against aggregation. Up-regulation of HSPB1 in injured sensory and motor neurons confirmed its important function in the survival of peripheral neurons [36,37]. Mutations in HSPB1, such as P182L disrupt the chaperone activity and promote the formation of protein aggregates [38,39]. In contrast, mutations such as R127W, S135F, and R136W endow HSPB1 with increased chaperone activity and enhanced binding to their client proteins [13], which seemingly prevent the formation of protein aggregates. However, these hyperactive mutations still lead to the degeneration of peripheral neurons, suggesting that the protein aggregates accumulate in neurons and their clearance is disturbed. 

The large protein aggregates generated in axons are predominantly eliminated by autophagy [40,41,42]. Under stress conditions, sequestration of cytotoxic misfolded proteins in the autophagic machineries, including aggresomes and autophagosomes, is vitally important for reducing stress damage [43]. Then, the toxic aggregates or autophagosomes travel a long distance in the axon to reach perinuclear lysosomes for degradation [44]. A pioneering study by Aplin et al. [45] shed light on the indispensable role of microtubules in the degradation of misfolded proteins. They found that the microtubule inhibitor nocodazole can block the fusion of autophagosomes with lysosomes, eventually leading to the accumulation of autophagic vacuoles. They proposed that a microtubule network was required to direct the movement of autophagosomes. Autophagosomes are motile structures, and their movement along intact microtubules and subsequent fusion with lysosomes is essential for protein degradation [46]. In addition, previous studies have found that toxic protein aggregates can form aggresomes at the perinuclear compartment, which are also degraded by autophagy pathway [42]. Inhibition of microtubule formation by nocodazole blocks the formation of aggresomes [47], prolonging the half-life of aggregated proteins [48]. These researches suggest that autophagic clearance of stress-induced misfolded proteins is heavily reliant on microtubule-based axonal transport. 

Axonal transport of organelles, lipids, proteins and misfolded proteins is essential for the survival and function of a neuron [49]. Microtubules are polar cylindrical polymers composed of α- and β-tubulin, with their stable minus-end (at which α-tubulin is exposed) toward the cell body and growing plus-end (at which β-tubulin is exposed) toward axon tips [21]. Microtubules in axonal transport serve as tracks along which various cargos such as autophagosomes and the protein aggregates are transported by motor proteins of dynein to the perinuclear region [22,50]. Thus, the retrograde transport (that is, toward the cell body) is vital for the clearance of misfolded proteins produced in axons to avoid the build-up of toxic protein aggregates [51]. Thus, impairment of microtubule-based axonal transport could lead to defects in autophagy, which has been identified as the underlying pathogenesis for several neurodegenerative disorders such as Alzheimer’s disease, amyotrophic lateral sclerosis, Huntington and Parkinson’s disease [52,53]. 

Disturbance of microtubule-based axonal transport was also linked to CMT symptoms [14]. Defective mitochondrial transport along microtubules was observed in DRG isolated from symptomatic HSPB1^S135F^-expressing mice, owing to the deacetylation of α-tubulin driven by enhanced recruitment of HDAC6. Acetylation of α-tubulin is required for long-lived microtubules, which confers the ability of long-range transport to neurons [54]. In DRG isolated from presymptomatic mice, however, both levels of α-tubulin acetylation and mitochondrial transport were unaffected by the HSPB1^S135F^ isoform [15]. This difference suggests that the reduction of mitochondrial transport is not the underlying cause of the degeneration of peripheral neurons and the pathology of CMT. As discussed by Almeida-Souza et al. [13], the excessive binding of HSPB1 mutants with client proteins may disturb their normal function essential for the maintenance and survival of peripheral neurons. Thus, we here focused on the biological effects of the aberrant binding of HSPB1^S135F^ to α-tubulin, particularly under repeated stress conditions. 

In the present work, we confirmed enhanced interactions of HSPB1^S135F^ with α-tubulin but also acetylated α-tubulin, which is consistent with the previous research [15]. We further confirmed that these interactions are not altered during repeated heat shock. We found that heat-shock stress triggers a large increase in the level of ubiquitylation associated with misfolded proteins. Our work further indicates that the abnormal binding of HSPB1^S135F^ to α-tubulin impeded the microtubule-based transport of ubiquitinated protein aggregates and autophagosomes along for the degradation by autophagy, resulting in the accumulation of ubiquitinated protein aggregates and further decrease of cell adaptability to repeated stress. Reduction in cell adaptability to cellular stress is potentially pathogenic because it lowers the threshold at which endogenous or exogenous noxious agents cause irreversible cellular damage [55]. Given progressive and selective degeneration of peripheral neurons in CMT patients, it can be deduced that decreased cell adaptability caused by the accumulation of stress-induced misfolded proteins may be an important contributor to axonal damage of peripheral neurons. Recently, HSPB1^S135F^ has been found to impair autophagic flux upon starvation [56]. The results presented here further suggest the link between selective degeneration of peripheral neurons and repeated stress that peripheral neurons encountered in daily life. In addition, the autophagy defect observed in this work may be different from that in aging brain diseases such as Parkinson’s disease, Tauopathy, and Alzheimer’s disease. The age-related protein aggregation in the latter is mainly caused by the disrupted regulation of protein homeostasis in aged neurons [57].

It must be noted here that although misfolded proteins and retrograde movement of mitochondria are both transported by dynein motor along microtubules, mitochondria utilizes different mechanisms to recruit dynein [50]. Furthermore, mitochondria is a special structure that can be transported toward both plus- and minus-end, and previous studies have proved the existence of mitochondria-specific transport regulation mechanisms in axons [58]. The elaborate regulation of mitochondria transportation might explain why mitochondrial transport was unaffected by HSPB1^S135F^ in DRG isolated from presymptomatic mice. For the symptomatic one, however, decreased acetylation of α-tubulin may be responsible for impaired mitochondria transport [14]. Acetylation of tubulin makes microtubule flexible, and its reduction may decrease microtubule adaptability to environmental stress such as mechanical force acting on peripheral neurons [59], further leading to microtubule breakage and thus impaired mitochondria transport [54]. Therefore, we will further investigate how HSPB1^S135F^ affects the response of peripheral neurons to mechanical force from daily exercise in the coming work, which will provide a complete understanding of the molecular aspects of the pathogenesis of CMT.

Conclusively, we proposed an integrative stress-based model for understanding the HSPB1^S135F^-induced accumulation of misfolded proteins, which may provide a novel insight into the pathogenesis of CMT. As shown in Figure 8, in HSPB1^WT^-expressing cells, stress-induced misfolded proteins can be elaborately regulated by protein quality control systems (PQC), including molecular chaperones, UPS and autophagy. In cells expressing HSPB1^S135F^, however, HSPB1^S135F^ binds to α-tubulin and thus blocks the transport of autophagosomes and protein aggregates along microtubules for autophagic degradation. As a result, stress-induced misfolded proteins accumulate at the cell edge, which decreases cell adaptability to repeated stress. Over time, cells that are susceptible to external stimuli will be damaged. Similarly, expression of HSPB1^S135F^ in peripheral neurons leads to defects in microtubule-based axonal transport under repeated stress, thus decreasing the clearance of protein aggregates. Without competent autophagy, accumulated misfolded proteins and protein aggregates will induce the degeneration of peripheral neurons. Our findings revealed for the first time that expression of HSPB1^S135F^ decreases cell adaptability to repeated stress, which lays an important theoretical foundation for further understanding the relationship between HSPB1 mutations and selective degeneration of peripheral neurons in CMT patients.

## Figures and Tables

**Figure 1 cells-11-02886-f001:**
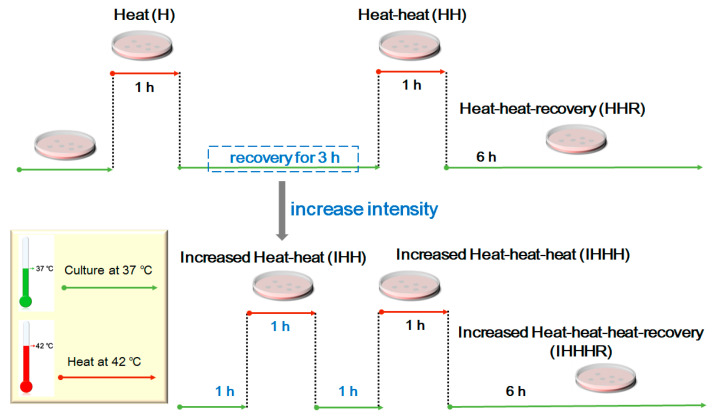
Schematic representation of heat-shock experimental design.

**Figure 2 cells-11-02886-f002:**
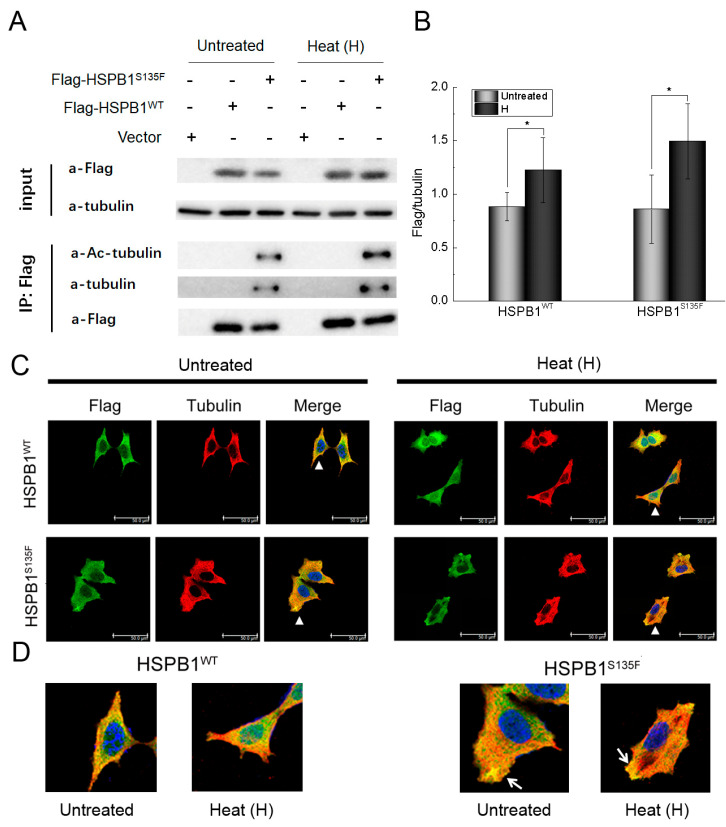
HSPB1^S135F^ binds to α-tubulin and acetylated α-tubulin. (**A**,**B**) Western blot and co-immunoprecipitation of HEK293T cells demonstrated that heat shock resulted in the increase of HSPB1^WT^ and HSPB1^S135F^ expression. HSPB1^S135F^ displayed binding affinity to α-tubulin and acetylated α-tubulin. HSPB1 was detected with the antibody of Flag tag; ***, *p* < 0.05 versus the respective untreated group. (**C**) Immunofluorescence staining of SH-SY5Y cells confirmed that HSPB1^S135F^ displayed increased co-localization with α-tubulin (Scale bars: 50 µm). For clarity, cells indicated by arrowheads in (**C**) were amplified and presented in (**D**), and co-localization of HSPB1^S135F^ and α-tubulin were pointed out by arrows.

**Figure 3 cells-11-02886-f003:**
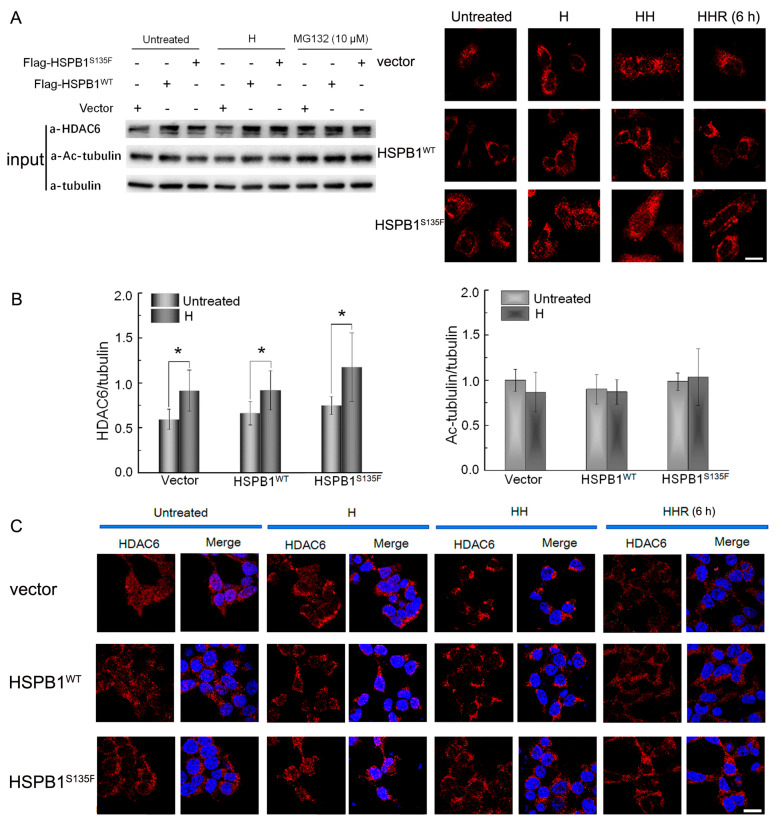
HSPB1^S135F^ prevents perinuclear translocation of HDAC6. (**A**) Western blot of HEK293T cells treated with repeated heat shock or MG132. (**B**) Quantification of HDAC6 and acetylation level of tubulin; ***, *p* < 0.05 versus respective untreated group. Intracellular distribution of HDAC6 (red) was investigated by immunofluorescence staining of HEK293T cells (**C**) and SH-SY5Y cells (**D**). Upon repeated heat shock, HDAC6 was found to agglomerate into punctate or linear aggregates in cells with vector and HSPB1^WT^ while into large diffused aggregates in cells with HSPB1^S135F^. Scale bars: 20 µm.

**Figure 4 cells-11-02886-f004:**
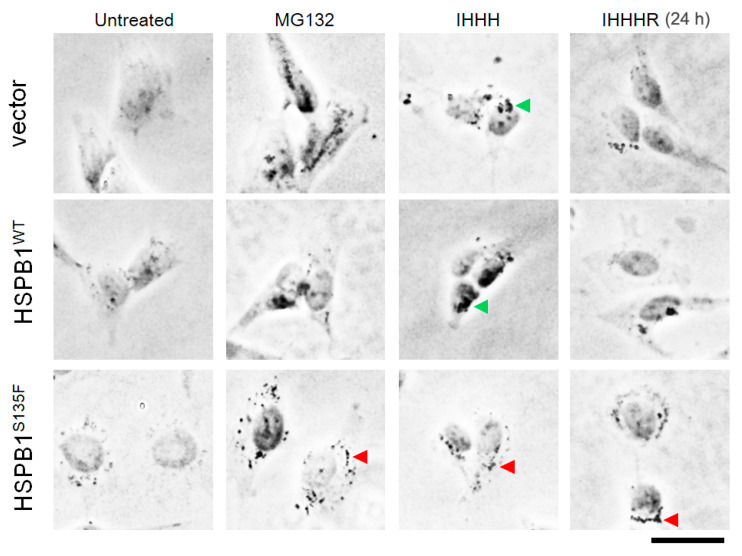
Aggresome formation in SH-SY5Y cells was probed by phase contrast microscope. Perinuclear aggresomes in cells with vector and HSPB1^WT^ are indicated by green arrowheads. Aggresomes were absent in HSPB1^S135F^ cells, no matter treated with MG132 or heat shock. Scatter black dots indicated by red arrowheads were not degraded in cells expressing HSPB1^S135F^ after recovery of 24 h. Scale bar: 50 µm.

**Figure 5 cells-11-02886-f005:**
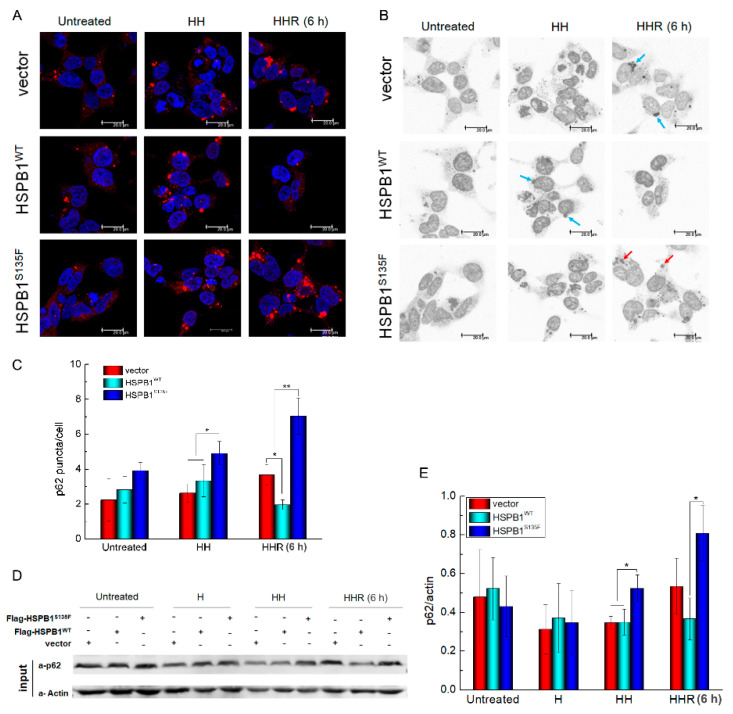
HSPB1^S135F^ prevents clearance of p62 punctuate inclusions. (**A**) Immunofluorescence imaging of p62 punctuate inclusions in HEK293T cells treated with heat shock or MG132. (**B**) Intracellular distribution of p62 punctuate inclusions in HEK293T cells. Perinuclear punctuate inclusions are shown by blue arrows; whereas cytoplasmic ones indicated by red arrows. Scale bars: 20 µm. (**C**) Average number of p62 punctuate inclusions per cell calculated by Image J; ****, *p* < 0.01. (**D**,**E**) Western blot of p62 expression in HEK293T cells; ***, *p* < 0.05.

**Figure 6 cells-11-02886-f006:**
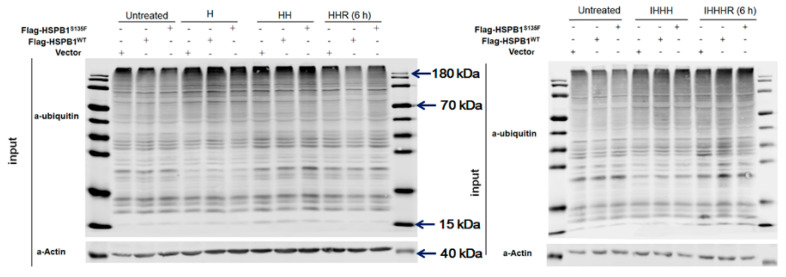
HSPB1^S135F^ prevents the clearance of ubiquitinated proteins in HEK293T cells.

**Figure 7 cells-11-02886-f007:**
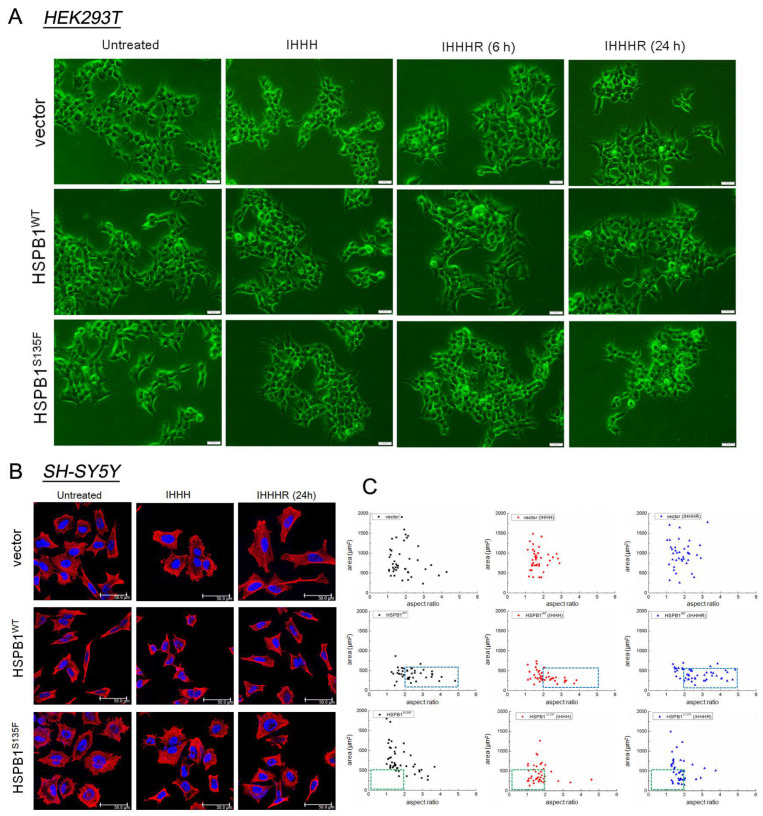
HSPB1^S135F^ decreases cell recovery from repeated heat shock. (**A**) Survival state of HEK293T cells treated with heat shock was observed by invert phase contrast microscope; scale bars: 50 µm. (**B**) Cell morphology of SH-SY5Y was investigated by immunofluorescence staining of tubulin (red) and nucleus (blue); scale bars: 50 µm. (**C**) Cell area and aspect ratio of SH-SY5Y cells were measured and calculated by Image J; *n* > 40 per group. The blue boxes show complete recovery of HSPB1^WT^-expressing cells from repeated heat shock, while the green boxes indicate that most of HSPB1^S135F^-expressing cells cannot recovery from the repeated heat shock.

**Figure 8 cells-11-02886-f008:**
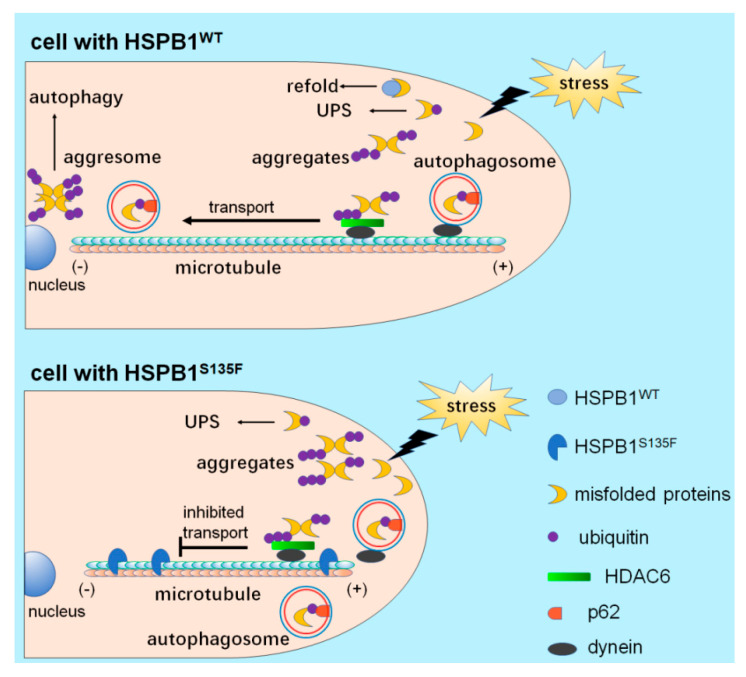
An integrative stress-based model of HSPB1^S135F^-induced accumulation of misfolded proteins.

## Data Availability

Data is contained within the article or Appendix A.

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
