# Peer review of "A Charcot-Marie-Tooth-Causing Mutation in HSPB1 Decreases Cell Adaptation to Repeated Stress by Disrupting Autophagic Clearance of Misfolded Proteins"

_cells, 2022, doi:10.3390/cells11182886_

Round 1
Reviewer 1 Report
Zhang et al. describe a set of experiments performed on HEK293T and SH-SY5Y cells in the attempt to elucidate the pathogenic mechanisms underlying CMT2F, an autosomal dominant neuropathy caused by mutations in the HSPB1 gene. In particular, they focus on the S135F variant, which is also associated with distal hereditary motor neuropathy, type IIB.
Cells are transfected to overexpress either the wild-type or the mutant protein and are subjected to a heat-shock based stress.
It is found that the mutant protein has an increased affinity to alpha-tubulin and this presumably impairs the clearance of misfolded protein compromising both the perinuclear translocation of aggresomes and the clearance of auophagosomes/lysosomes. Overall, cells expressing the mutated protein do not recover after stressful events.
Since similar alterations of the microtubule-based axonal transport are associated to neurodegenerative diseases such as CMT itself, Alzheimer, Parkinson, Huntington, and ALS, the authors propose that CMT2F pathogenesis is related to the incapacity of PNS cells to recover from stressful events that cause the accumulation of misfolded proteins.
Major issues.
The authors are presenting as they own consideration which have been published between 2015 and 2020. A quick pubmed search reveals that some of the experiments and many results here reported are already published.
Although, the authors are adding some new data the failure to correctly cite the literature and discuss the results of the scientific community make this work unacceptable both from a scientific and an ethical point of view.
I strongly suggest reading the following manuscripts which must have been pivotal both for the experimental set-up and for the discussion of the results.
1) PMID: 30669930 - Neuropathy-causing mutations in HSPB1 impair autophagy by disturbing the formation of SQSTM1/p62 bodies.
In this work, HeLA and SH-SY5Y cells overexpressing the S135F mutation and stressed under starvation showed a decrease in the autophagic flux. This is in accordance with published work on other small HSP. Specifically, it is found that "... the ability of SQSTM1 to form SQSTM1/p62 bodies is impaired by neuropathy-causing mutations in HSPB1, which display a decreased formation of phagophores upon starvation." and the authors conclude that "By confirming the deleterious effect of HSPB1 mutants in patient-derived motor neurons, our study presents strong evidence of autophagy impairment as a pathomechanism of HSPB1-linked neuropathies."
2) PMID: 26675522 - Truncated HSPB1 causes axonal neuropathy and impairs tolerance to unfolded protein stress
In this work, it is shown that fibroblasts from patients carrying the R127L and Met169Cfs2* mutations are less tolerant to heat-stress and that Met169Cfs2* increase sensitivity to protein misfolding.
3) PMID: 32323160 - Small heat shock proteins in neurodegenerative diseases
This work recapitulates the state of the art of HSPB1 involvement in neurodegenerative diseases.
4) PMID: 29330367 - Characterization of human small heat shock protein HSPB1 α-crystallin domain localized mutants associated with hereditary motor neuron diseases
In this paper it is suggested "that the disease-related mutations of HSPB1 [including S135F] modify its self-assembly and interaction with partner proteins thus affecting normal functioning of HSPB1 in the cell"
Another critical point, unexplained in the manuscript is on which basis the authors hypothesized an interaction of alfa-tubulin with HSPB1.
The idea of stress intolerance + accumulation of misfolded protein as key pathogenetic events in CMT2F is not new; the results reported only confirm this mechanism.
The manuscript must be completely rewritten, taking in consideration all available literature data which must be discussed with results.
MINOR POINTS
A) In the introduction, CMT2F (along with its OMIM number) should be cited the first time HSPB1 mutations are mentioned
B) Material and Methods. It is not clear how the control experiments were set up and if technical/biological replicates were made.
C) English needs to be revised throughout the paper. As an example, it is "nucleus" and not "nuclear"; "punctuate" is a verb and terms such as "punctate inclusions" or "punctate structures" should be preferred.
Reviewer 2 Report
The mutation of heat-shock protein B1 (HSPB1) has been known in Charcot-Marie-Tooth (CMT) disease. This study investigated the relationship between HSPB1S135F, tubulin and autophagy after heat-shocked and double heat-shocked condition.
1. The manuscript format should be checked.
2. Why heat-shocked and repeated heat-shocked were applied to mimic neurodegenerative CMT disease?
3. In vivo animal study or primary culture neuron is suggested since the authors described the HSPB1S135F-expressing mice in the introduction.
4. SH-SY5Y neuroblastoma cell line was used in this study. It may not be suitable to represent SH-SY5Y cells as neurons which the authors showed a pathogenic model of peripheral neurons in Fig. 8 according to their results.
5. What is the meaning of HSPB1S135F from cytosol to nucleus in Fig. 2?
6. Line 212, “These results implied that the main function of heat shock-induced HDAC6 was to handle misfolded proteins.” Why could the authors conclude to this result?
7. The authors site a 2003 paper mentioned that aggresomes can easily been observed under a phase-contrast microscope. However, many markers and detectors have been reported since 2017. The authors should double confirm the aggresomes.
8. Why green and blue arrows were used to indicate perinuclear punctuates in Fig. 5B? Moreover, the blot of p62 did not match to the quantification in Fig. 6D and E.
Reviewer 3 Report
Congratulations. It is a great job! I just recommend a grammar revision.
Round 2
Reviewer 1 Report
Zhang and collaborators carefully considered the comments and suggestions proposed and submitted a revised version of their manuscript. Both Introduction and Discussion now include up-to-date literature on HSPB1 and CMT-related pathogenetic mechanisms.
The manuscript is delivering some interesting messages:
1) it confirms that impaired autophagy is connected to peripheral neuropathy/neurodegeneration
2) it provides a link between HSPB1 mutations, alpha-tubulin and impaired autophagy
3) it report testing the effects of repeated heat shock-based stress on a common HSPB1 variant (S135F), in contrast to starvation that was the main stressor in previously published papers
On the other hand, the authors should be more careful on their conclusions which should still be considered in the context of available literature or simply toned down:
The sentence "Conclusively, we proposed a stress-based model for understanding the HSPB1-S135F-induced accumulation of misfolded proteins, which may provide a novel insight into the pathogenesis of CMT" should be rephrased considering that the authors are supporting data available in the literature, not proposing novel models.
Also sentences such as "Our findings revealed for the first time that expression of HSPB1S135F decreases cell adaptability to repeated stress" are considered “priority claims”. Again, I suggest to rephrase.
The authors should revised their discussion according to these principles.
I again want to highlight what is already known in the literature:
1) The accumulation of misfolded proteins in HSPB1-S135F is already reported (see for instance PMID: 31366698 or "Numerous stressors, including hyperthermia and hypoxia, can induce the expression of Hsps, which, in turn, interact with client proteins and co-chaperones to regulate cell growth and survival" in PMID: 30483047).
2) Misfolded proteins, in turn, cause a stress of the Endoplasmic Reticulum and misfolded proteins are either re-folded or degraded through autophagy (see for instance PMID: 26389781).
3) Since HSPB1 mutations impair autophagy (another known fact) it is not strictly "novel" that stressing a cell carrying a mutation that alters autophagy leads to a compromised function.
Overall, the results of this paper could therefore be seen as the confirmation and an important extension of an already known pathogenic mechanism of peripheral neuropathy based on an unprecedented stressor (heat-shock, which triggers the action of heat-shock proteins).
Other points:
-The reason why heat-shock mimics the daily insults received by the PNS could have been explained better. Were the authors thinking about high fevers, which may exacerbate a peripheral neuropathy, or simply the difference in heat between seasons or due to physical exercise?
-Previously published works on HSPB1-S135F show a decrease in acetylated alpha-tubulin that is not present here. How do the authors discuss this difference?
-The "perspective for targeting autophagy as a promising therapeutic strategy" mentioned in the abstract is another well-established knowledge, as for instance in PMID: 29130394 - "This information, therefore, suggests that the possibility of targeting autophagy may represent a potential approach to improve conditions of affected individuals".
There is also a funded project: "Treatment of autophagy deficits in Charcot-Marie-Tooth disease caused by mutations in the small heat shock proteins HSPB1 and HSPB8." on https://researchportal.be/en/project/treatment-autophagy-deficits-charcot-marie-tooth-disease-caused-mutations-small-heat-shock
- Figure legends are inadequate and some panels are difficult to interpret without an extensive knowledge of the experimental procedure (see, for instance, Figure 3C).
- An interesting perspective as to why nerves are more susceptible to damage than other cells is proposed by Haidar and Timmerman, PMID: 28553203:
"The morphological features that distinguish neurons from other cells are their post-mitotic nature, highly polarized structure, and extended cytoplasm into axons and dendrites that can stretch far from the cell body. The latter feature is more exaggerated in neurons supplying the peripheral nerves. The spatial compartmentalization of neurons makes them prone to aggregation and accumulation of dead organelles and misfolded proteins. Autophagy therefore forms an essential homeostatic process for neurons. Knocking out key autophagy genes, such as ATG7 in mouse neurons leads to neurodegeneration (Komatsu et al., 2006). Defective autophagy has also been linked with numerous neurodegenerative diseases (Frake et al., 2015). The high susceptibility of neurons to autophagic impairment could explain why mutations in ubiquitously expressed genes can cause neuron-specific pathology in inherited neuropathies."
Reviewer 2 Report
Ø For Point 2, is there any reference to support the description?
